# How does policy modelling work in practice? A global analysis on the use of epidemiological modelling in health crises

Liza Hadley[1,2]*, Caylyn Rich[3], Alex Tasker[4], Olivier Restif[1], Sebastian Funk[2]

**1** Department of Veterinary Medicine, University of Cambridge, Cambridge, United Kingdom, **2** Centre for the Mathematical Modelling of Infectious Diseases, London School of Hygiene and Tropical Medicine, London, United Kingdom, **3** Colorado Health Outcomes Center, University of Colorado Anschutz Medical Campus, Denver, Colorado, United States of America, **4** University of Bristol, Bristol, United Kingdom

* LH667@CAM.AC.UK

## Abstract

This study examines the use and translation of epidemiological modelling by policy and decision makers in response to the COVID-19 outbreak. Prior to COVID-19, there was little readiness for global health systems, and many science-policy networks were assembled ad-hoc. Moreover, in the field of epidemiological modelling, one with significant sudden influence, there is still no international guidance or standard of practice on how modelled evidence should guide policy during major health crises. Here we use a multi-country case study on the use of epidemiological modelling in emergency COVID-19 response, to examine the effective integration of crisis science and policy in different countries. We investigated COVID-19 modelling-policy systems and practices in 13 countries, spanning all six UN geographic regions. Data collection took the form of expert interviews with a range of national policy/ decision makers, scientific advisors, and modellers. We examined the current use of epidemiological modelling, introduced a classification framework for outbreak modelling and policy on which best practice can be structured, and provided preliminary recommendations for future practice. Full analysis and interpretation of the breadth of interview responses is presented, providing evidence for the current and future use of modelling in disease outbreaks. We found that interviewees in countries with a similar size and type of modelling infrastructure, and similar level of government interaction with modelling reported similar experiences and recommendations on using modelling in outbreak response. From this, we introduced a helpful grouping of country experience upon which a tailored future best practice could be structured. We concluded the article by outlining context-specific activities that modellers and policy actors could consider implementing in their own countries. This article serves as a first evidence base for the current use of modelling in a recent major health crisis and provides a robust framework for developing epidemiological modelling-to-policy best practice.

**Data availability statement:** All relevant data are within the paper and its Supporting Information files.

**Funding:** LH acknowledges support from the Wellcome Trust (block grant no. RG92770). SF also acknowledges support from the Wellcome Trust (grant no. 210758/Z/18/Z). The funders had no role in study design, data collection and analysis, decision to publish, or preparation of the manuscript.

**Competing interests:** The authors have declared that no competing interests exist.

# 1 Introduction

Epidemiological modelling was brought to the forefront of public health decision-making during the COVID-19 pandemic, in many countries for the first time. Epidemiological modelling is a branch of epidemiology that uses mathematics to predict the spread of disease through a population [1]. It is especially useful for estimating the impact of possible prevention and control measures. Many countries relied heavily on this type of scientific advice for resource allocation, planning, and forecasting during the pandemic, especially in the face of unprecedented public health interventions and incomplete knowledge [2–7]. It is highly likely that epidemiological modelling will continue to be needed in future disease control, due to the complex and uncertain nature of both pathogen spread and population behaviour [1,8–13]. The uptake of modelling, and more broadly science advice as a whole, in policy can be streamlined with improvements to the organisation, translation, and timeliness of scientific evidence. Previous works have identified many open questions [14,15], and it has become clear that best practices are needed for navigating the new prominence of modelling advice in public health decision making. Consequently, our study addresses the *organisation* and *translation* of epidemiological modelling in policy spheres. We used COVID-19 epidemiology as a case study for health crisis systems, and encourage broader public health researchers to build off this work for navigating best practice discussions with their own governments.

As countries have invariably faced differing challenges in COVID-19, a cross-country comparison on the use and organisation of epidemiological modelling in policy is vital. It is essential to document these practices such that we can learn from shared experiences. Of most interest are the perceptions of those who were *directly* involved in COVID-19 modelling response - national epidemiological modellers and the policy/ decision makers (government employees) and scientific advisors that they interacted with. (See the principal eligibility criteria in S1 Text for an explanation of these roles). Particular academic groups have reflected and reported on their own work (for example Howerton et al. in the US, Sherratt et al. in the UK [16,17]) but there is strong need for a breadth study drawing together global practices.

Dedicated research studies on the use and organisation of modelling in outbreak response have previously been sparse. Emergent COVID-19 reflective studies include those run by the interagency COVID-19 Multi-model Comparison Collaboration (CMCC), the KEMRI-Wellcome Trust, and Imperial College London among others [4,18–21]. In 2020, the CMCC produced early guidance on how models can be used in COVID-19, and set out three distinct classes of policy questions suitable for modelling (projecting epidemiological impact in the absence of intervention; scenario analyses and impact of interventions; modelling related to costs) [19]. The group also discussed presentation of results and gave brief examples of the modelling-to-policy infrastructure and collaborations in a few select countries. Further work is now needed to examine modelling-to-policy infrastructures and communication pathways in detail, to provide an evidence base of shared experiences on the integration of scientific advice in health crises.

Our own study builds on this research gap. We examined the use of epidemiological modelling in COVID-19 response in 13 different countries and jurisdictions. Our study aims to:

(a) Compare national practices on the use and organisation of epidemiological modelling in COVID-19 decision-making (3.1-3.5);

(b) Identify contextual drivers linking the shared experiences of interviewees and develop a classification framework to structure future guidance (3.2);

(c) Propose preliminary recommendations on the use of epidemiological modelling in health policy, contextual to the different country settings identified in (b) (3.6).

An adjacent paper Hadley et al., 2025 reports findings on the topic of communication and visualisation [22].

## 2 Methods

This Methods section follows the consolidated criteria for reporting qualitative studies (COREQ) [23]. Details on ethics approval (2.1), study design (2.2), country and participant selection (2.3), interview format and location (2.4), data collection and analysis (2.5), and inclusivity (2.6) are provided in the following subsections. Additional information about the study is included in S1 Text and in Hadley et al., 2025 (thesis forthcoming) [24].

### 2.1 Ethics statement

The University of Cambridge Psychology Research Ethics Committee (application number PRE.2023.034) has reviewed this research. The majority of study participants did not know the interviewer prior to study commencement but all participants were informed of the purpose of the study, reasons for interest, and proposed outputs at the recruitment stage. All participants provided informed written consent.

### 2.2 Study design

This is a qualitative descriptive study. Our study aimed to compare national practices on the use of epidemiological modelling in COVID-19 decision-making, identify key contexts linking the shared experiences of interviewees to develop a classification framework, and propose preliminary recommendations for future practice.

Data collection took the form of in-depth semi-structured interviews with national COVID-19 policy and decision-makers, science advisors, and epidemiological modellers in 13 different countries and jurisdictions, spanning all six UN geographic regions. Data analysis used thematic analysis and ideal-type analysis. Note that our paper is descriptive, aiming to describe the integration of epidemiological modelling in current health policy systems and advance a typology framework and recommendations for future use. It is not an explanatory study providing for example reasoning for COVID-19 outcomes in different countries.

### 2.3 Country and participant selection

This study defines epidemiological modelling activity in a broad sense, to include statistical modelling and analytics too. In total, 12 countries and 1 jurisdiction (Hong Kong, China) took part in the study. For brevity, we use the term 'country' to mean 'country or jurisdiction' throughout this article, with the knowledge that Hong Kong, China is a jurisdiction or 'special administrative region' and not itself an independent country. Country identification was performed using a targeted selection process; the lead researchers first carried out an informal scoping review of the use of epidemiological modelling in national COVID-19 responses to assemble a longlist of candidate countries for further exploration. Countries with a population of less than 5 million were excluded. Fifteen target countries were chosen to support a target sample size of 20–30 interviewees following current practices in similar qualitative interview research studies [25]. Data saturation was

not a primary goal as this study was intended to be exploratory. Candidate countries were identified using intersectional selection criteria: covering each UN geographic region, low-and-middle-income countries and high-income countries, low- and high- COVID-19 prevalence, and diversity of prior experience of modelling for national decision-making, modelling structures or consortiums (including those with no formal capacity), pre-COVID-19 pandemic preparedness ratings, and population size. Researchers then used purposeful sampling to contact key modelling-to-policy actors in each country by email for interview [26]. If no relevant participants could be sought, a second country was chosen until the study met the desired study size of 10–15 countries with 2+participants per country, including at least one country from each UN geographic region. In total, 27 participants from 13 countries took part. 31 participants replied to the study recruitment invite but three were unavailable at the time of interview and one could not be taken forward as a second interviewee from the same country could not be found. Recruitment ran from 21st April 2023–10th May 2024.

Participant names are kept anonymous but participants reported the following as their countries (or jurisdictions) of practice: Australia, Canada, Colombia, France, Japan, Hong Kong Special Administrative Region of the People's Republic of China (Hong Kong), Kenya, New Zealand, Peru, Republic of Korea (South Korea), South Africa, Uganda, and the United Kingdom. At least two respondents were recruited from each of the above countries. Table 1 describes the countries in our study and Table 2 describes the make-up of participants. Of the 27 interviewees, 12 self-identified as scientific advisors or policy/ decision makers (3, 9 respectively), and 15 as modellers. Where possible, efforts were made to make sure both modellers and science advisors or policy/ decision makers were interviewed in each country. By country, the split of interviewees achieved was fairly even: the majority of countries in our study (8/13) had both a modeller and a science advisor or policy/ decision maker as interviewees; but it is noted that in three countries, only modellers were interviewed and in the remaining two countries, only scientific advisors or policy/ decision makers were interviewed. Interviewees self-identifying as scientific advisors and policy/ decision makers included national Government Chief Scientific Advisors and heads of the relevant national Health department or agency, as well as intermediaries or convenors assigned specifically to coordinate or relay modelling activities.

**Table 1. The 13 countries and jurisdictions involved in our study and relevant demographic statistics.**

| Country of practice | Country details | | | |
|---|---|---|---|---|
| | UN geographic region[a] | Income status[b] | GHS pandemic preparedness rating (2019)[c] | Population size (millions)[d] |
| Australia | Oceania | HIC | 4th | 25.8 |
| Canada | Northern America | HIC | 5th | 38.0 |
| Colombia | LAC | UMIC | 65th | 51.2 |
| France | Europe | HIC | 11th | 64.5 |
| Hong Kong, China | Asia | HIC | 51st* | 7.5 |
| Japan | Asia | HIC | 21st | 124.9 |
| Kenya | Africa | LMIC | 55th | 52.5 |
| New Zealand | Oceania | HIC | 35th | 5.1 |
| Peru | LAC | UMIC | 49th | 33.5 |
| South Korea | Asia | HIC | 9th | 51.8 |
| South Africa | Africa | UMIC | 34th | 59.1 |
| Uganda | Africa | LIC | 63rd | 45.1 |
| United Kingdom | Europe | HIC | 2nd | 67.2 |

*Each interviewee reported their country(ies) of practice as one of the above. LAC = Latin America and the Caribbean; LIC = Low-income country; LMIC = Lower-middle income country; UMIC = Upper-middle income country; HIC = High-income country. Country data is taken from the following sources: (a), (b) & (d) UN geographic region, country income, and population (as of 1st January 2021) is as described in the United Nations World Population Prospects 2022 [27,28]; (c) Rankings taken from the 2019 Global Health Security Index, indicating expected overall preparedness for an infectious disease outbreak [29]. Note that Hong Kong's ranking of 51st represents the overall ranking for China - as noted in the main text, Hong Kong is a Special Administrative Region of China.*

In Table 1, we see that the 13 countries in our study had a diversity of income status (low, lower-middle, upper-middle, and high), pre-pandemic preparedness ratings (ranging from 2nd to 65th), and population size (ranging from 5.1 million to 124.9 million). However it is noted that additional data on the peculiarities of each country such as public-private partnerships and multisectoral vs multidisciplinary activity was not readily available. Readers interested in public-private partnerships are directed to related study Dabak et al. (in preparation) [30].

## 2.4 Interview format and location

Interviews were in-depth and semi-structured, lasting approximately 1 hour each. Author LH (female PhD student with training and prior experience in qualitative interviewing) carried out all interviews. The format was structured discussion on four topics in turn, facilitated by the interviewer. Additionally, the first interviewee in each country was invited to co-create a 'sketch diagram' with the interviewer mapping out the key actors in modelling and policy in their country from their perspective. This rough diagram served as a reference point for discussion, and the main pathways were verbally verified through discussion with subsequent interviewees. The remainder of the interview followed the standard semi-structured format.

The interview schedule was heavily informed by previous work on 'Challenges for Future Pandemics' in 2020/2021 [15]. Hadley et al., 2021 introduce a succinct framework that defines the areas for progress on the interface of outbreak modelling and policy (reproduced here as Fig 1). This idea-generating piece directly informed our study's core topics and specific interview questions. Hadley et al., 2021 identified seven broad challenges of pandemic modelling for policy; we narrowed this down to focus on four related interview topics, namely 'Structures and pathways to policy' (relating to challenges 1 & 3 in Fig 1), 'Collaboration and knowledge transfer' (2&4), 'Communication and visualisation' (5), and 'Evaluation and reflection' discussing recommendations for future practice (7). Our study did not discuss technical model challenges, as this has been frequently researched and reported on in the wider literature (as one example, see Sherratt et al. discussing European nations [31]). The full interview schedule is included in S1 Text and questions were piloted on three colleagues before study commencement.

Four interviews were conducted in-person in private workplace meeting rooms; the remaining 23 were conducted online via video-conferencing software. Majority of interviews were conducted one-to-one in English, with one using an independent translator to aid communication.

## 2.5 Data collection and analysis

All interviews were audio recorded, professionally transcribed, qualitatively coded, and then reviewed for accuracy by the study researchers. Analysis of data followed a two-pronged approach: using thematic analysis for descriptive findings and ideal-type analysis for a more in-depth understanding of the data, enabling contextual recommendations to be made on the future use of modelling [32,33]. Thematic analysis is the main method for identifying, analysing, and reporting patterns within qualitative interview data, and is best used when the research aim is to explore people's beliefs, knowledge, or experiences from a set of qualitative data such as interview transcripts [32,34]; hence it was deemed an appropriate method of choice for our study. Our thematic analysis followed a deductive approach, utilising topics posited in earlier

**Table 2. Make-up of study participants.**

| Affiliation | Role | No. of interviewees |
|---|---|---|
| Academic | Modeller | 15 |
| | Independent scientist and advisor | 3 |
| Government | Policymaker | 5 |
| | Senior government (For example head of a Ministry of Health or Public Health Agency, senior Chief Scientific Advisors, senior members of Cabinet, and related roles) | 4 |

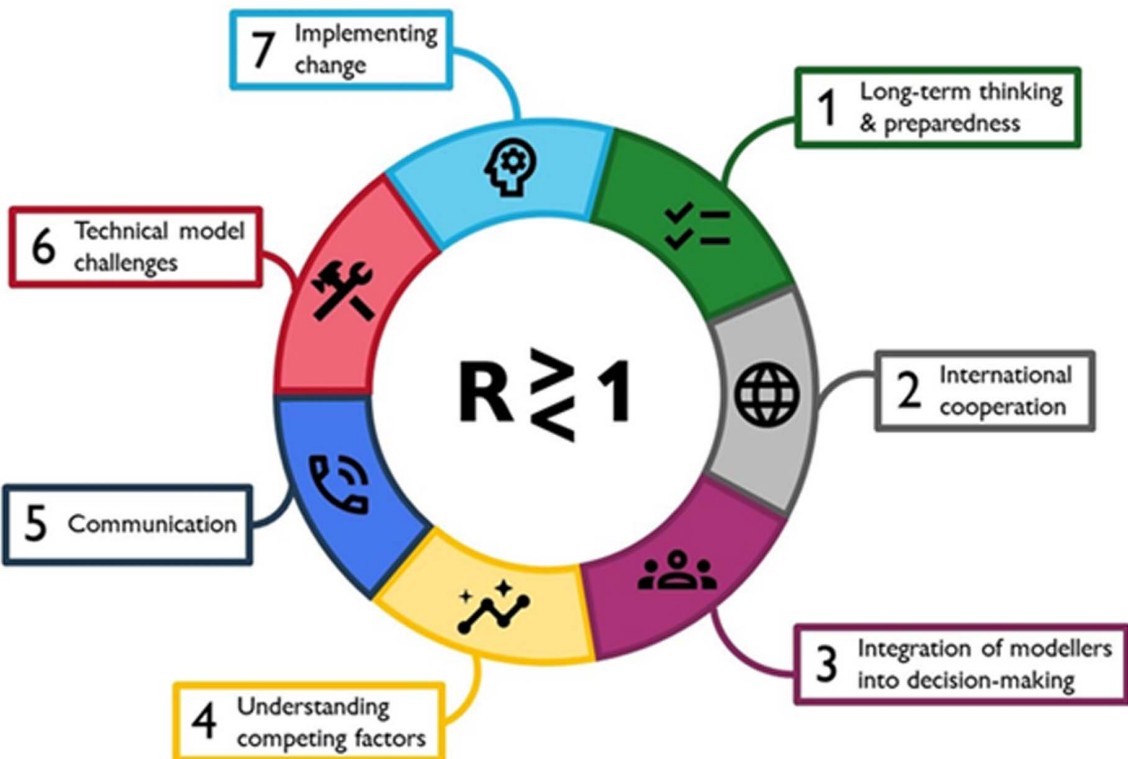

**Fig 1. Infographic depicting seven broad challenges of pandemic modelling for policy, around the central equation of epidemic spread.** Reproduced from Hadley et al., 2021 [15].

study Hadley et al., 2021 (described in Section 2.4 above). Two of the researchers (LH, CR) led the initial coding with manual line-by-line review. Manual coding was chosen over the use of software to enable a deeper understanding of the material. The first six transcripts were double-coded and then compared to reach a consensus on the coding framework. Remaining transcripts were then coded and independently reviewed to enable triangulation and continued code consistency. The researchers also produced analytic memos for each interview. Initial codes were then merged to account for duplication, from which meaningful basic themes and organising themes were derived, reviewed, and refined [35]. The final step of thematic analysis was interpretation and reporting of the major findings.

Ideal-type analysis was carried out by researchers LH and AT, following a modified protocol adapted from Stapley et al. [33]. Ideal-type analysis is a method for systematically developing a typology (a useful categorisation) of the subject of interest [33,36,37]. This analytical approach was selected to organise interview responses and to understand aspects of an interviewee's experiences that could contribute to shared beliefs and shape recommendations for future practices in outbreak modelling and policy. Guided by the question "What are the key contexts to consider when comparing shared experiences and recommendations on the integration of outbreak modelling in policy?", we sought to understand contextual drivers that would enable the development of tailored guidance for best practice to move beyond the current approach of blanket recommendations which are rarely contextualised. While typologies are commonly employed in qualitative data analysis, there is no agreed best practice for constructing them [38]. Given the subject and design of our study we chose to follow the method of Stapley et al. to identify key characteristics influencing the subject of interest (beliefs and recommendations on future practices), collecting examples of optimal cases, and defining contexts within which shared experiences were observed (forming a description of each 'ideal type') [33]. Ideal types were tested, refined, and checked for

credibility, before using a structured analytical approach to organise into a classification framework highlighting important similarities and differences between groups. The results of the classification framework are presented in section 3.2 (Fig 3) and are used to inform the preliminary contextual recommendations given in Table 5. Further detail on the derivation of this is provided in the Results and in S1 Text.

Once data analysis was complete, a draft version of this article was also circulated to all interviewees to enable verification and opportunity for revisions of the key findings.

Results sections 3.1 and 3.2 present key data from each country and introduce a classification framework for important contexts to consider in outbreak modelling and policy. Section 3.3 examines any collaboration of modelling with other scientific disciplines, section 3.4 explores translators and knowledge brokers, and section 3.5 discusses mistakes. Finally, section 3.6 draws on the complete study analysis, supplementary results in S1 Text, and pre-existing literature to introduce guidelines for future practice on using epidemiological modelling in health crises. Results on communication and visual preferences are presented in our adjacent study Hadley et al., 2025, entitled "Visual preferences for communicating modelling: a global analysis of COVID-19 policy and decision makers" [22].

### 2.6 Inclusivity in global research

Additional information regarding the ethical, cultural, and scientific considerations specific to inclusivity in global research is included in S2 Text.

## 3 Results

### Factors driving shared experiences

Thematic and ideal-type analysis identified two driving parameters linking the shared experiences of interviewees: size/type of modelling infrastructure, and level of government interaction with modelling. For example, clusters of interviewees with similar reported experiences, beliefs, and recommendations on the integration of outbreak modelling in policy had a similar type and size of modelling infrastructure and similar level of government interaction with modelling. This was evidenced by interviewees' reported strengths and challenges of modelling-policy systems which commonly framed as relating to either a large multi-team modelling capacity, limited (single-team) modelling capacity, or disjointed modelling capacity; and contextual to modellers being either embedded within government or independent. These structural factors were a strong driver for similarities between groups of interviewees than for example views on communication and visualisation where recommendations were more universal across interviewees. These structural factors were also a stronger driver than countries with the same income status, and countries in the same region. Findings on type of modelling infrastructure and level of government interaction are presented in section 3.1, and analysis of contextual drivers linking shared experiences in outbreak modelling and policy is presented in section 3.2.

### 3.1 Modelling infrastructure and pathways to policy

We observed four distinct types of national **modelling infrastructure**, presented in Fig 2. These are: one modelling team; multiple small teams functioning as one; a consortium consisting of multiple teams and multiple models led by a modelling committee; and finally, teams working in isolation, feeding independently into government. Types 2 and 3 were separated to distinguish a single-model infrastructure that would likely have no capacity to work on the same questions with a multi-model infrastructure that does have capacity for self-verification, reproduction, or consensus (in any form). Type 4 considers settings where modelling activity is disjoint; modellers likely do not confer with each other, and results are instead 'combined' in-house by a government non-modelling committee. Examples are given in Fig 2. Our classification aims to capture the size of each country's primary modelling capacity, ability or inability to reproduce model findings, and whether

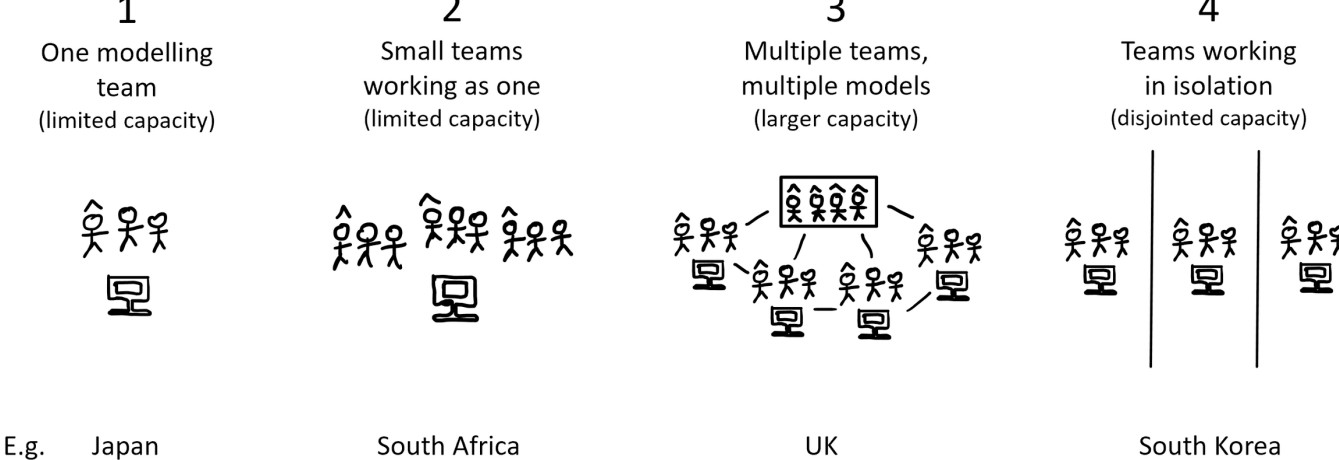

| 1 | 2 | 3 | 4 |
|---|---|---|---|
| One modelling team (limited capacity) | Small teams working as one (limited capacity) | Multiple teams, multiple models (larger capacity) | Teams working in isolation (disjointed capacity) |

E.g.　　Japan　　　　　　　　South Africa　　　　　　　　UK　　　　　　　　South Korea

**Fig 2. *The four types of national modelling infrastructure observed in our study.*** One modelling team (1); multiple small teams functioning as one (no capacity to reproduce) (2); multiple teams and multiple models (capacity to reproduce) led by a committee of modellers (3); modellers and/or modelling teams do not collaborate, work in isolation, and feed results directly into government to be combined by a committee of non-modellers (4). Pictured: Individuals with hats represent modelling principal investigators/ senior modellers. Computers represent models. Boxes represent a modelling committee. For example, in South Africa (type 2), there was insufficient capacity to examine the same modelling questions with multiple models so teams divided the research questions among themselves and functioned as one team. In South Korea (type 4), modelling effort was disjointed - modelling teams did not collaborate or discuss findings and instead sent reports directly to the relevant policy actor.

modelling outputs are *synthesised and interpreted by modellers* (1–3) or *non-modellers* (4) such as other infectious diseases or policy experts.

Additionally, one can classify each country by the **level of government interaction in modelling**. This is shown in Table 3 (rightmost column), along with summary findings on the structures and pathways to policy in each country. The leftmost columns concern *modelling* infrastructure while the rightmost columns consider the *policy infrastructure for modelling* and provide a relative ranking for the level of government interaction and independence in each setting, based on in-depth discussions with interviewees. One sees for example that the level of interaction in modelling from government bodies varied greatly (Medium: 3 countries; High: 5; Very high: 2; Embedded: 2). Illustrative definitions of these rankings are provided in S1 Text. Note that countries where modelling had little to no use by national government in COVID-19 were excluded from our study.

### 3.2 Classification framework for outbreak modelling and policy

Identifying countries with similar experience on the use of modelling will enable the development of more effective recommendations. As noted in the 'Factors driving shared experiences' subsection, ideal-type analysis suggests that the size/ type of modelling capacity and level of government interaction with modelling were contextual drivers linking the shared experiences, beliefs, and recommendations of subsets of interviewees. These are important contexts to consider when grouping country settings by their current use of outbreak modelling in policy, to produce meaningful further guidance. Section 3.1 identifies four types of national modelling infrastructure (Fig 2) and four levels of government interaction with modelling (Table 3), creating sixteen possible categories of shared experience. Upon systematically comparing and contrasting interview transcripts and summaries, this was reduced to five key categories of outbreak modelling and policy (Fig 3). See Methods section 2.5 and S1 Text for a description of this analysis.

The following five categories of modelling-policy systems were identified as having similar experiences and recommendations for future practice: (A) Countries with small modelling capacity and likely high government linkage; (B) Countries

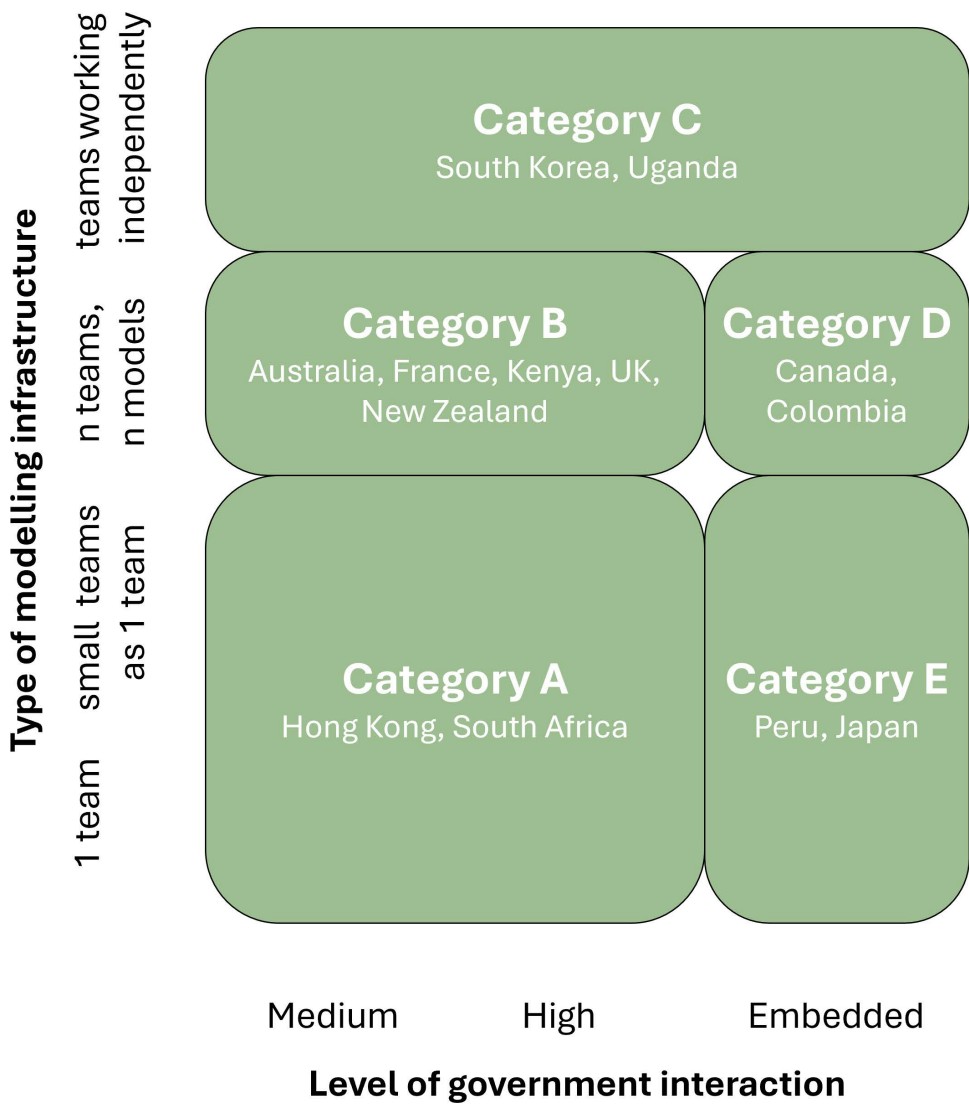

**Fig 3. Classification framework for outbreak modelling and policy.** Five categories of modelling-policy systems were identified and organised by similar experiences and recommendations on the use of outbreak modelling in policy. Type of modelling infrastructure and level of government interaction with modelling were identified as important contextual drivers when grouping country settings. The five categories of modelling-policy systems identified are: (A) Countries with small modelling capacity and likely high government linkage, such as Hong Kong; (B) Countries with large modelling capacity and high or very high government linkage, such as Kenya; (C) Countries where modelling teams worked in isolation, combined by non-modellers in government, such as South Korea; (D) Countries with a large modelling capacity with a primary government modelling team, such as Canada; (E) Countries with one government modelling team, such as Peru. Countries were placed in the same category if interviewees demonstrated similar beliefs and experiences on the use of outbreak modelling in policy. Note: For other pathogens and other health crises, each category may contain different countries.

with large modelling capacity and high or very-high government linkage; (C) Countries where modelling teams worked independently, combined by non-modellers in government; (D) Countries with a large modelling capacity with a primary government modelling team; (E) Countries with one government modelling team (Fig 3). This is called a typology in qualitative research [33].

Producing a robust categorisation of shared experiences, beliefs, and recommendations in outbreak modelling and policy as shown in Fig 3 facilitates context-specific recommendations for future practices. To this end, subsequent results are presented by these categories where it is meaningful to do so.

**Table 3. Modelling infrastructure and details on the structures and pathways to policy in each country in COVID-19, as described by interviewees.**

| Country | Type of modelling infrastructure (type #) | Name of MC or team if given | Main affiliation of modellers | Main government linkage point(s) | Level of (structural) government interaction |
|---|---|---|---|---|---|
| *Africa* | | | | | |
| Kenya | **Multiple teams, multiple models (3)** | Led by Centre for Epidemiological Modelling and Analysis (CEMA), University of Nairobi | Universities or research institutes | Health, also Education, Faith *Multiple linkages reported: MoH mainly, also Ministry of Education, National Council of Churches and religious groups, National Taskforce, and Cabinet at times* | **High.** Kenyan COVID-19 modelling review team sits within the National COVID-19 Task Force. |
| South Africa | **Small teams working as one (2)** | South African COVID-19 Modelling Consortium (SACMC) | Universities or research institutes | Health, Economics *Ministry of Health advisory groups, MoH direct at times, Treasury* | **High.** SACMC convened by the National Institute of Communicable Diseases at the request of MoH. |
| Uganda | **Teams working in isolation (4)** | NA | Universities or research institutes | Health *Epidemiological Advisory Group, PRESIDE and National Planning Authority* | **Medium.** Two modellers on relevant SACs. |
| *Asia* | | | | | |
| Hong Kong, China | **One team (1)** | Led by LKS Faculty of Medicine, University of Hong Kong researchers | Universities or research institutes | Health *COVID-19 Expert Advisory Panel, and Cabinet at times* | **High.** Highly-integrated key individuals as brokers. |
| Japan | **One team (1)** | Led by Kyoto University researchers | Universities or research institutes | Health *Ministry of Health* | **Very high.** Modellers in-housed to MoH during first wave and modeller on relevant SAC. |
| South Korea | **Teams working in isolation (4)** | NA | Universities or research institutes | Health *Korean Disease Control and Prevention Agency (CDC)* | **Medium.** Brokerage carried out by influential ID specialist due to limited modelling expertise. |
| *Europe* | | | | | |
| France | **One team** initially (1); then **multiple teams, multiple models (3)** | Led by Institut Pasteur researchers; later the Coordinated Action of Modelling for Disease (ACM) | Universities or research institutes | Health *Conseil Scientifique* | **Medium.** One modeller on national SAC throughout. |
| United Kingdom | **Multiple teams, multiple models (3)** | Scientific Pandemic Influenza Group on Modelling, Operational (SPI-M-O) | Universities or research institutes primarily, with government contributors | Health primarily *SAGE (Government Office for Science); Cabinet Office at times* arily | **High.** Several modellers on national SAC throughout, secondments to Cabinet Office, and regular attendees from Cabinet Office to MC meetings. |
| *Latin America and the Caribbean* | | | | | |
| Colombia | **Multiple teams, multiple models (3)** | Led by National Institute of Health (INS) | Government, with university/ research institute teams contributing on occasion | Health, Education *Ministry of Health, Ministry of Education* | **Embedded** (primary modelling team). |
| Peru | **One team (1)** | CDC-Peru modelling unit | Government | Health *Ministry of Health* | **Embedded.** |
| *Northern America* | | | | | |

*(Continued)*

**Table 3.** (Continued)

| Country | Type of modelling infrastructure (type #) | Name of MC or team if given | Main affiliation of modellers | Main government linkage point(s) | Level of (structural) government interaction |
|---|---|---|---|---|---|
| Canada | **Multiple teams, multiple models (3)** | Led by Public Health Agency of Canada (PHAC) | Government, with university/ research institute teams contributing regularly | Health *Minister of Health, Cabinet* | **Embedded** (primary modelling team). |
| *Oceania* | | | | | |
| Australia | **One team** (scenarios) (1); **multiple teams, multiple models** (forecasting) (3) | Doherty Modelling consortium and National COVID-19 Situational Assessment Consortium | Universities or research institutes primarily | Health, Economics *Multiple linkages including Australian Health Protection Principal Committee (AHPPC), Treasury* | **High.** Long-term pre-existing relationship and modellers on multiple SACs. |
| New Zealand | **Multiple teams, multiple models (3)** | COVID-19 Modelling Aotearoa (CMA) | Universities or research institutes | Health *Ministry of Health, Department of Prime Minister and Cabinet* | **Very high.** Extremely close science-policy system but still with modellers' independence. |

*MC = Modelling Committee. SAC = A multidisciplinary Scientific Advisory Committee or expert panel. MoH = Ministry of Health/ Department of Health. ID = Infectious Diseases. Illustrative definitions used to identify level of government interaction with modelling are shown in S1 Text.*

### 3.3 Collaboration with other disciplines

In the majority of cases, modelling activity was focussed on primary health outcomes (i.e., COVID-19 transmission and disease) but there were a few interdisciplinary success stories where modellers collaborated with other disciplines. Costing, or the idea of evaluating trade-offs (either health or economic based) was reported by modellers in New Zealand, Australia, Canada, and Peru, but was explicitly indicated as not existent in the UK, Colombia, and France. In New Zealand, while economic modelling remained the responsibility of the Treasury, the country's modelling consortium (COVID-19 Modelling Aotearoa, CMA) collaborated to provide analyses of isolation policy trade-offs and linked informal health modelling within the CMA to economic modelling within the Treasury.

On the behavioural sciences front, in Australia modellers previously collaborated with social scientists on the use of citizen juries to inform ethical vaccine allocation, but during the COVID-19 response, such networks suffered a lack of behavioural social sciences inclusion. A respondent from South Korea also emphasised the importance of an interdisciplinary approach to modelling: "This is a policy where extensive educational, social, and economic losses are expected, so it is necessary to listen to the opinions of experts in each field." Different forms of science can of course be combined 'higher up' in senior government, but there was a shared sentiment of desire for more cross-pollination in countries that lacked it. For example, post pandemic, modellers from Kenya reported that they are now actively working with behavioural scientists in a local School of Anthropology on the current Rift Valley Fever outbreak.

Note: Limitations to the scope of this analysis resulted in a lack of evaluation for the *quality* of collaboration across disciplines, but rather sought to determine whether collaboration existed.

### 3.4 Translators and knowledge brokers

We define 'translators' as those individuals who took on the role of packaging information, converting complex epidemiological modelling findings into meaningful language for policy. In countries where modellers collaborated and had a level of independence from government (categories A and B in Fig 3), this 'translator' role was often taken on by the most senior

modeller(s) (Australia, France, Kenya, South Africa) or epidemiologist (Hong Kong). Exceptions include New Zealand's small close-knit science-policy ecosystem where modellers interacted directly with senior government and first translation was performed largely by the relevant Chief Scientific Advisor, and the UK whose modelling committee (SPI-M-O) was jointly convened by a policy co-chair and academic co-chair. In the latter, the policy co-chair and SPI-M-O secretariat were all civil servants. Uniquely these UK individuals had significant past modelling expertise and could effectively synthesise key modelling findings that the policy co-chair, crucially *inside* government, could take into the parts of government that the academics reportedly had limited access to. Secretariat members ensured that an accurate interpretation of the modelling was relayed to ministers by meticulously agreeing appropriate slide headings with Cabinet Office officials to ensure the knowledge was conveyed appropriately; the benchmark became whether policy makers would be able to understand the implications of the content if they only read the headings.

Countries with more government embeddedness (categories D and E in Fig 3) saw a mix of translators: the Chief Public Health Officer in Canada, the lead academic modeller in Japan, and the relevant Peru CDC department director in Peru. Finally, in South Korea (category C - disjointed modelling) where there were reported to be no dedicated infectious disease modelling teams at the beginning of the COVID-19 pandemic (primarily mathematics or public health research groups with only some modelling expertise), translation was taken on by an experienced and informed non-modelling infectious diseases principal investigator. The origin of each of the above interactions varied; as one interviewee

**Table 4. Responses to the question "What was your biggest mistake (in relation to the use of modelling)?". All responses have been paraphrased to maintain anonymity.**

| What was your biggest mistake (in relation to the use of modelling)? |
| --- |
| Not being transparent enough with lack of knowledge; allowing policy makers to espouse more certainty in the models than existed |
| Inability to address vaccine hesitancy - lacked incorporating behavioural or economic scientists into modelling which contributed to elevated vaccine hesitancy |
| Underestimating the number of deaths |
| Relying on a single model/ not managing to develop a multiple model consortium |
| Lack of understanding of types of modelling and modelling's limitations (not investing in the pedagogy of modelling) |
| Not taking an interdisciplinary approach, only collaborating with the Ministry of Health rather than other ministries |
| Utilising worst-case scenarios to justify actions, especially when presented in a public forum |
| Erring too far on the pessimistic side of modelling/ the negative outcome or worst-case scenario |
| Accidentally getting too involved with politically motivated media figures |
| Lack of resources (financial) to pool expertise |
| Risk in using models as the primary reason for introducing a lockdown |
| Not believing vaccines would be available so quickly |
| Not grasping the impact of interventions |
| Lack of resourcing creating biases in datasets - marginalised communities couldn't enjoy same level of for example surveillance consequently forcing 'national' rather than 'local' models onto indigenous communities |
| Some modellers/ modelling groups struggled to recognise the time and place appropriate for complex vs simple models |
| Narrow scope of metrics, no modelling of socioeconomic impacts |
| Downplaying models' predictions |
| "I don't think we made a mistake but with hindsight we've always thought, thank God, we kind of did it right and we didn't get it wrong" |
| Not being prepared capacity-wise |
| Poorly communicating the risks |

commented "it is hard to create strict rules around who should communicate: it ends up being the persons involved where their strengths lie and what structures they have around them".

## 3.5  Mistakes

Many respondents were asked what they perceived to be the 'biggest mistake' when using modelling in the response. Responses are shown in Table 4 for interest. These comments were shared with the interviewer in the spirit of identifying shared challenges and should be interpreted not as shortcomings of a particular individual but as areas where we could collectively improve.

## 3.6  Recommendations for future practice

Towards the end of each interview, interviewees discussed their key takeaways and recommendations for the future of outbreak modelling and policy (presented in full in S1 Text as 'Results on recommendations for future practice (from interviewees)'). This dataset highlighted four essentials for successful outbreak modelling and policy: data, systems, communication, and relationships (Fig 4).

Expanding on recommendations from interviewees, the results of our study analysis, and additional literature in this area, we, the authors, introduce preliminary guidelines for future practice on using epidemiological modelling in disease outbreaks, shown in Table 5.

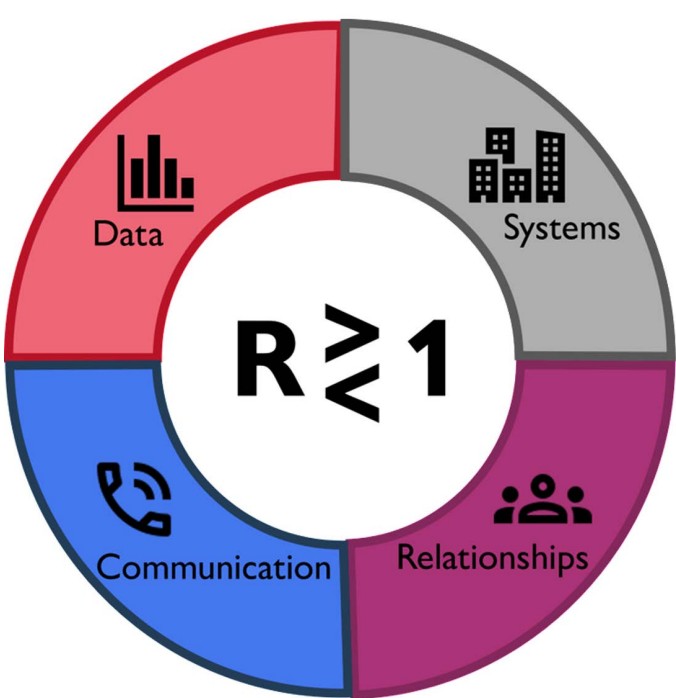

**Fig 4. Essentials for successful outbreak modelling and policy - data, systems, communication, and relationships.** This figure follows the style of earlier work Hadley et al, 2021.

**Table 5. Preliminary guidelines for future practice on using epidemiological modelling in disease outbreaks. From the results of our study analysis, recommendations from interviewees, and additional literature in this area, we suggest the following guidelines for future practice on using epidemiological modelling in disease outbreaks.**

| | Guidelines for future practice on using epidemiological modelling in disease outbreaks | |
|---|---|---|
| 1 | Identify your system | There is no optimal modelling-policy system. Moving between categories of modelling-policy systems may require significant political and cultural change. Modellers should instead identify the system in which they operate (Fig 3) and utilise the context-specific case studies and guidance presented in this article. |
| 2 | Relationships | Facilitate long-standing relationships. Consider developing relationships with government groups rather than individuals, to preserve future linkages under government turnover. For systems where modelling sits primarily 'outside' government, it is critical to have an interpreter/translator or advocate (who may or not be a modeller) within policymaking spheres to provide a space for modelling. |
| 3 | Standing capacity for modelling | More standing capacity for modelling is needed in some countries. For systems with limited modelling capacity, provide secure jobs for modellers. Disjointed modelling effort is encouraged to unify informally. Where there is a lack of capacity to train people in-country, an international modelling network or society could aid development. Large modelling capacity and multi-model consortiums will likely need to be maintained with dedicated funding; where that exists we encourage active sharing of ideas to countries with fewer experts. |
| 4 | Data availability | Strong data access and availability is essential for modelling. This is a universal recommendation for all settings. |
| 5 | Legitimacy | For settings where modelling sits 'outside' of government, establish relationships and agreements/ memorandums of understanding ahead of time. Unifying modelling effort through a modelling committee can aid identifiability in policy spheres. Retrospective verification of modelling results is also important for developing trust and legitimacy. |
| 6 | Communicating modelling | Caution against overbelieved or misinterpreted modelling. For detailed guidance on communicating and interpreting modelling in policymaking, see adjacent study [22]. |
| 7 | Interdisciplinarity | Science advice in health crises should be holistic, looking beyond direct health outcomes. However, research on how modelling can best interact with economics and behavioural sciences is limited. We offer examples (Section 3.3) but further research is needed before a formal guideline can be made here. |

## 4 Discussion

At the time of shock, when the pandemic first hit, nations rapidly organised scientific and modelling capacity, and crucially this self-organisation happened in *different* ways (Table 3), resulting in *different successes and challenges* as outlined in detail above. Countries with a similar size and type of modelling infrastructure, and similar level of government interaction with modelling were found to have reported similar successes and challenges on using modelling in their COVID-19 response (section 3 'Factors driving shared experiences'). More universally, there was a desire for epidemiological modelling to have an eye to wider societal impacts, and in many settings, work formally with other experts. We highlight two success stories of epidemiological modelling successfully working with economics and social science either in the COVID-19 pandemic (New Zealand) or as an active post-COVID-19 advancement (Kenya) (section 3.3).

On the communication front, we introduced the idea of *re-translation*, acknowledging the need to ensure that not only is modelling translated and packaged effectively to policy/ decision makers, but that in subsequent iterations 'up the chain' the message is maintained. Further results on communication and visualisation from this study are presented in the adjacent paper Hadley et al. [22]. There is a need now for further emphasis in Science Communication work on enabling *subsequent* translation of modelling, at least in policy settings.

The classification framework for outbreak modelling and policy introduced in Fig 3 highlights key contextual considerations (specifically the size and type of modelling capacity, and independence from government or embeddedness) across which groups of interviewees reported similar experiences, beliefs, and recommendations. It is important to note here that interviewees in the same region or in countries with the same income status did not necessarily report similar experiences in COVID-19: For example, within the region of Africa, Kenya benefitted from a large multi-team modelling consortium while South Africa and Uganda had more limited modelling capacity (Table 3). In the region of Asia, modellers in Japan reportedly had a very high level of integration with the government through the Ministry of Health while in South Korea, efforts were disjointed and due to insufficient principal investigators in epidemiological modelling, communication was

taken on by an infectious diseases specialist (Section 3.4). One can see that simply grouping countries by income status or region would not accurately capture the nuance of reported experience in COVID-19 epidemiological modelling and policy. Instead, having a single-team, multi-team, or disjointed modelling capacity, and being independent from or embedded within government processes did well-represent the different experiences and recommendations of the study group. The classification framework introduced in Fig 3 hence provides a useful basis for wider examination of the use and translation of epidemiological modelling in outbreaks. There is an opportunity for future work here to test our classification framework with countries beyond the 13 considered, with a larger sample size, and in other (for example routine public health) disease settings.

Finally, we observed overarching themes of data, communication, systems, and relationships as primary enablers to the effective use of epidemiological modelling in disease outbreaks. We synthesised specific and contextual recommendations for future practice on the use of modelling in health crises in the 13 countries examined. Table 5 outlines guidance for the future use of epidemiological modelling in disease outbreaks. These stem from crucial suggestions from key global actors, many with extensive experience in multiple health events and from different perspectives.

Looking at related literature in this area, Owek et al. identified pre-existing relationships between researchers and policy/ decision-makers as an enabling factor for successful knowledge translation, in addition to co-creation of advice and embeddedness [4]. Our findings echo this, with 'Systems' and 'Relationships' both being pivotal for future outbreak modelling and response. Our findings on systems also reflect that of formal inquiries taken on by several countries - for example in the UK's COVID-19 Inquiry, modellers (as well as all other scientists, policy/ decision makers, and associated actors) provided witness statements detailing their involvement in the country's pandemic response from which it is clear that the systems and relationships built prior to and during COVID-19 were fundamental to modelling's utility in COVID-19 [39].

Additionally there was acknowledgement of the role of translators by the CMCC report discussed earlier in this article, noting that where needed "decision makers may be supported by advisors or knowledge brokers with a high literacy to interpret models" [19]. In contrast, Owek et al. found that face-to-face debriefings between scientists and policy/ decision makers were preferred in Low and Middle Income Countries over the use of a knowledge broker or intermediary [4]. We observed more of a mix of translators in our study - of the five countries classified as LMICs in Table 1 (Colombia, Kenya, Peru, Uganda, and South Africa), translation was taken on by the most senior modeller in three settings but by an intermediary in the other two. It is worth acknowledging whether academics should be the ones responsible for translating results into policy language or if this activity could be more effectively in-housed. There is merit for the latter in the countries we have investigated.

In terms of limitations, we have aimed to capture a range of experiences in our study but there was a limited number of participants (27) and in most cases, only two participants per country. Consequently, we did not attribute recommendations to specific countries (with the exception of giving examples) and instead presented the major analyses by typology (e.g., suggestions for countries with small modelling capacity, suggestions for countries where modelling sits 'outside' of government, etc.). Note also that interview data represents the *perceptions* of interviewees, hence this study only evaluates the perceived use of modelling, albeit from highly-informed individuals. Nonetheless, this may not reflect official policy or how individuals would behave in practice. We also use the terms 'government' and 'policy/ decision makers' loosely throughout, and acknowledge that this relates to different individuals and committees in each country, due to the multifaceted political structures. A closer examination of these roles is a point for future work [40]. The wider political, economic, and cultural factors experienced by different government actors may also influence the uptake of scientific evidence, but this was outside the scope of our article.

We acknowledge that state-level and regional (multi-country) modelling were outside the scope of this study, as were case study countries where modelling *did not* contribute to national COVID-19 response. State-level modelling played a role in some of the larger countries of our study, such as Colombia, Canada, and Australia, but these activities are not reported here. Examining countries with little or no use of modelling in outbreaks is suggested for further research as this could inform key challenges or recommendations that have not yet been reported.

 

## Conclusions

With the above limitations in mind, our study provides a first evidence base for the use of epidemiological modelling in different countries during a major recent health crisis (COVID-19). Analysis has enabled a classification framework and context-specific recommendations to be suggested (Fig 3, Table 5), from which further research and testing can be built.

Looking to the future, we advocate for actors to identify the system they are in and look to other countries reported in this article with a similar setup (similar size and organisation of modelling capacity, similar level of government involvement - Fig 3) for key actions to consider for strengthening the use of epidemiological modelling in outbreaks. No optimal modelling-policy system was observed - there were benefits to both small and large modelling capacity and to in-house vs independent modelling capacity. However, disjointed modelling capacity may be ineffective, with isolated modellers in Uganda and South Korea struggling to identify and recruit adequate support for modelling or networks within the discipline. Data in this article and in the supplementary dataset (contained in S1 Text) can inform how countries similar to yours have managed data sharing challenges, communication, relationships, etc., as many of the same challenges were observed by interviewees in similar systems in our study (Section 3.2). Although Fig 3 places countries in one of five categories based on COVID-19, we note that infrastructure can change significantly over time, across different disease expertise, or in non-crisis times, so it will be important to update these classifications. For example, post-COVID-19, some of the countries in our study (for example Canada, Kenya, South Korea) reported that they have been able to maintain or adapt their COVID-19 modelling-policy systems post-pandemic, but at the time of interview, New Zealand and South Africa had been unsuccessful in sourcing long-term funding for modelling-policy systems and their formal epidemiological modelling committees has been discontinued (S1 Text). In addition, Australia has since begun developing a Centre for Disease Control to maintain the relationships built between communicable disease scientists and actors in COVID-19 [41].

From the guidance given in Table 5, we also advocate for strong data access as a universal recommendation. For example, the Peru Ministry of Health and Peruvian CDC are working to improve their data sources after citing poor intervention data as a significant barrier in COVID-19 modelling and outbreak response (S4). For countries with limited epidemiological modelling capacity, for example a single team with little capacity for self-verification, reproduction, or consensus with multiple models, there is a critical need to build standing capacity through secure jobs, training opportunities, and where possible support from international modelling networks or societies. For example, mass training has taken place in Latin America and the Caribbean through an open online epidemiological modelling training course held fully in Spanish by and for Latin American actors, after identifying a lack of capacity in modelling in this region [42].

And finally for countries where modelling is independent from government, it is critical to have an advocate within policymaking spheres to provide a space for modelling, such as was the case with South Africa's convenor or the UK's policy co-chair for COVID-19 modelling groups. These individuals enabled modelling to be integrated and interpreted correctly. Developing relationships with government groups rather than individuals and establishing agreements and memorandums of understanding ahead of time will also protect against high turnover and enable rapid scale-up in times of crises.

The relative peacetime that some countries are now in offers a unique opportunity for actors to evaluate and improve their epidemiological modelling-policy systems, making use of the ideas and context-specific case studies presented in articles like this one.

This study has directly contributed evidence to formal activities to develop best practice, including the Lancet Commission on 'Strengthening the Use of Epidemiological Modelling of Emerging and Pandemic Infectious Diseases' [43].

## Supporting information

**S1 Text. Supplementary material.**
(DOCX)

**S2 Text. Inclusivity in global research.**
(DOCX)

   

## Acknowledgments

The authors thank Sungmok Jung, Mircea Sofonea, and all other interviewees for dedicating their time and experience to the study.

## Author contributions

**Conceptualization:** Liza Hadley, Alex Tasker, Olivier Restif, Sebastian Funk.

**Data curation:** Liza Hadley.

**Formal analysis:** Liza Hadley, Caylyn Rich, Alex Tasker.

**Investigation:** Liza Hadley.

**Methodology:** Liza Hadley, Alex Tasker, Sebastian Funk.

**Supervision:** Alex Tasker, Olivier Restif, Sebastian Funk.

**Validation:** Liza Hadley.

**Writing – original draft:** Liza Hadley.

**Writing – review & editing:** Liza Hadley, Caylyn Rich, Alex Tasker, Olivier Restif, Sebastian Funk.

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
