## [Decision Letter · Decision Letter 0]

10 Mar 2025

PGPH-D-25-00077

How does policy modelling work in practice? A global analysis on the use of epidemiological modelling in health crises

Dear Dr. Hadley,

Thank you for submitting your manuscript to PLOS Global Public Health. After careful consideration, we feel that it has merit but does not fully meet PLOS Global Public Health’s publication criteria as it currently stands. Therefore, we invite you to submit a revised version of the manuscript that addresses the points raised during the review process.

Academic Editor Guillaume Fontaine: Thank you for your submission to PLOS Global Public Health.

In addition to the reviewers suggestions, please address these points:

1. Please rework the first paragraph of the Introduction so it does not start with "This study examines the organisation of epidemiological modelling in Covid-19 response in 13 different

47 countries and jurisdictions." This sentence could be moved to the paragraph starting with "Our own study builds on this research gap..."

2. The section "Data collection takes the form of in-depth semi-structured interviews with national decision-makers, science advisors, and epidemiological modellers in 13 different countries. Interviewees span all six UN geographic regions, and capture the breadth of Covid-19 experience." belongs to the methods.

3. At the start of the Methods section, please add a "Study design" subheading and specify the design of the study. Did you use a qualitative descriptive design?

4. Since this is a qualitative study, it would be worth reviewing the key points outlined in the Consolidated criteria for reporting qualitative research (COREQ) and make sure these are covered in the study. https://www.equator-network.org/reporting-guidelines/coreq/

We look forward to receiving your revised manuscript.

Kind regards,

Guillaume Fontaine, PhD, RN

Academic Editor

Journal Requirements:

 1. Please include a complete copy of PLOS’ questionnaire on inclusivity in global research in your revised manuscript. Our policy for research in this area aims to improve transparency in the reporting of research performed outside of researchers’ own country or community. The policy applies to researchers who have travelled to a different country to conduct research, research with Indigenous populations or their lands, and research on cultural artefacts. The questionnaire can also be requested at the journal’s discretion for any other submissions, even if these conditions are not met.  Please find more information on the policy and a link to download a blank copy of the questionnaire here: https://journals.plos.org/globalpublichealth/s/best-practices-in-research-reporting. Please upload a completed version of your questionnaire as Supporting Information when you resubmit your manuscript.  2. We note that your Data Availability Statement is currently as follows: “The available anonymised data has been presented in this manuscript and any associated manuscripts where possible.”  Please confirm at this time whether or not your submission contains all raw data required to replicate the results of your study. Authors must share the “minimal data set” for their submission. PLOS defines the minimal data set to consist of the data required to replicate all study findings reported in the article, as well as related metadata and methods (https://journals.plos.org/plosone/s/data-availability#loc-minimal-data-set-definition).  For example, authors should submit the following data:  - The values behind the means, standard deviations and other measures reported; - The values used to build graphs; - The points extracted from images for analysis.  Authors do not need to submit their entire data set if only a portion of the data was used in the reported study.  If your submission does not contain these data, please either upload them as Supporting Information files or deposit them to a stable, public repository and provide us with the relevant URLs, DOIs, or accession numbers. For a list of recommended repositories, please see https://journals.plos.org/plosone/s/recommended-repositories.  If there are ethical or legal restrictions on sharing a de-identified data set, please explain them in detail (e.g., data contain potentially sensitive information, data are owned by a third-party organization, etc.) and who has imposed them (e.g., an ethics committee). Please also provide contact information for a data access committee, ethics committee, or other institutional body to which data requests may be sent. If data are owned by a third party, please indicate how others may request data access.

Additional Editor Comments (if provided):

N/A

Reviewers' comments:

Reviewer's Responses to Questions

**Comments to the Author**

1. Does this manuscript meet PLOS Global Public Health’s publication criteria ? Is the manuscript technically sound, and do the data support the conclusions? The manuscript must describe methodologically and ethically rigorous research with conclusions that are appropriately drawn based on the data presented.

Reviewer #1: Partly

Reviewer #2: Yes

2. Has the statistical analysis been performed appropriately and rigorously?

Reviewer #1: N/A

Reviewer #2: Yes

3. Have the authors made all data underlying the findings in their manuscript fully available (please refer to the Data Availability Statement at the start of the manuscript PDF file)?

Reviewer #1: No

Reviewer #2: Yes

4. Is the manuscript presented in an intelligible fashion and written in standard English?

Reviewer #1: Yes

Reviewer #2: Yes

5. Review Comments to the Author

Reviewer #1: The manuscript addresses an important and timely topic concerning the role of epidemiological modelling in shaping public health decisions during crises. The study is well-structured, systematically conducted, and presents valuable insights from a diverse range of global experiences. The qualitative approach is appropriate given the study’s objectives, and the findings contribute significantly to ongoing discussions about best practices for integrating modelling into policymaking. However, there are areas requiring improvement in methodological clarity, analytical depth, and interpretation of findings.

Major Concerns

1. While the study employs a qualitative approach, the manuscript lacks sufficient justification for key methodological choices.

2. The authors mention stratified random sampling for the first country in each region, followed by purposive sampling. However, more details on how this selection was operationalised are needed.

3. It is unclear whether participants had equal representation across the different countries or whether biases exist in the selection of decision-makers versus modellers.

4. The authors mention thematic and ideal-type analysis but do not provide sufficient detail on the coding framework or how themes were derived.

The authors should provide more transparency on sampling and participant selection criteria and elaborate on the coding process and analytical approach.

5. The classification framework for modelling-policy systems is a strength, but its development lacks clear grounding in established typology methodologies.

6. The five identified categories appear descriptive rather than analytically derived. How were the key distinguishing factors determined?

The authors should justify the classification framework with references to typology literature and provide explicit criteria used to assign countries to each category.

7. The authors make broad claims about the implications of their findings for global health crises. However, given the study's sample size (27 participants), caution is needed in extending conclusions beyond the selected countries. The exclusion of countries where modelling was not integrated into COVID-19 response introduces potential selection bias, as lessons could also be drawn from those settings.

It will be prudent for the authors to acknowledge the limitations of sample size and selection bias and discuss the extent to which findings can be applied to other settings.

Minor Concerns

8. The manuscript inconsistently uses terms like “modellers,” “scientific advisors,” and “decision-makers.” A clearer distinction of roles would improve comprehension. Also, the term “modelling-to-policy systems” is introduced but not consistently used throughout.

9. The manuscript presents policy uptake of modelling as primarily a structural issue but does not sufficiently explore political, economic, or cultural factors that may influence decision-making.

10. The guidelines for future practice are useful but could be strengthened with more actionable steps.

11. The recommendation on interdisciplinarity acknowledges the need for collaboration with economics and behavioural sciences but does not suggest concrete strategies for achieving this.

12. Figure R1 (modelling infrastructure classification) is useful but would benefit from clearer labels distinguishing between different types of modelling structures.

13. Table R1 (summary of country systems) is informative but dense. A more visual representation (e.g., colour coding or summary statistics) could improve readability.

Reviewer #2: The manuscript titled "How does policy modelling work in practice? A global analysis

the use of epidemiological modelling in health crises" is original and first of its kind in the spectrum of epidemiological modelling analysis during global crises, with the COVID-19 pandemic being the case study. Although the introduction was precise and concise, it did not fully address the modelling justification for future occurrences. There is a need for clarity of the peculiarities of each country in each global geographical region in terms of categories of Loe-, middle- and high-income; multidisciplinary/multisectoral, hybrid versus streamlined discipline; and government involvement, Public-Private Partnership and private (institutional) approaches in the methods. The results were appropriate, and the discussion was also relevant. Still, more comparisons were required for aspects conducted during the non-crisis period in some parts of the geographical regions, like Europe and Asia. This is a magnificent manuscript that could be accepted for publication if all the aforementioned concerns were addressed at the end of the day.

6. PLOS authors have the option to publish the peer review history of their article (what does this mean? ). If published, this will include your full peer review and any attached files.

**Do you want your identity to be public for this peer review?** For information about this choice, including consent withdrawal, please see our Privacy Policy .

Reviewer #1: No

Reviewer #2: **Yes: ** Sikiru Olanrewaju Badaru

---

## [Decision Letter · Decision Letter 1]

2 May 2025

How does policy modelling work in practice? A global analysis on the use of epidemiological modelling in health crises

PGPH-D-25-00077R1

Dear Dr Hadley,

We are pleased to inform you that your manuscript 'How does policy modelling work in practice? A global analysis on the use of epidemiological modelling in health crises' has been provisionally accepted for publication in PLOS Global Public Health.

Best regards,

Julia Robinson

Executive Editor

Reviewer Comments (if any, and for reference):

Reviewer's Responses to Questions

**Comments to the Author**

1. If the authors have adequately addressed your comments raised in a previous round of review and you feel that this manuscript is now acceptable for publication, you may indicate that here to bypass the “Comments to the Author” section, enter your conflict of interest statement in the “Confidential to Editor” section, and submit your "Accept" recommendation.

Reviewer #1: All comments have been addressed

Reviewer #2: All comments have been addressed

2. Does this manuscript meet PLOS Global Public Health’s publication criteria ? Is the manuscript technically sound, and do the data support the conclusions? The manuscript must describe methodologically and ethically rigorous research with conclusions that are appropriately drawn based on the data presented.

Reviewer #1: Yes

Reviewer #2: Yes

3. Has the statistical analysis been performed appropriately and rigorously?

Reviewer #1: N/A

Reviewer #2: Yes

4. Have the authors made all data underlying the findings in their manuscript fully available (please refer to the Data Availability Statement at the start of the manuscript PDF file)?

Reviewer #1: Yes

Reviewer #2: Yes

5. Is the manuscript presented in an intelligible fashion and written in standard English?

Reviewer #1: Yes

Reviewer #2: Yes

6. Review Comments to the Author

Reviewer #1: This revised manuscript is well-organised, thorough, and well-written. The authors have successfully responded to earlier reviewer feedback and significantly strengthened the manuscript by clarifying their methodological rigour, enriching the typology with illustrative country data, and presenting recommendations that are appropriately contextualised.

The use of a qualitative descriptive design, supported by COREQ guidelines, is appropriate and systematically executed. The multi-country perspective enriches the value of the study, and the introduction of a classification framework (Figure R2) is both novel and useful for application in policy settings. The authors’ structured thematic and ideal-type analysis ensures that conclusions are well-supported by the data.

I appreciate the addition of country-specific detail in Tables R1 and M1/M2, which makes the findings more transparent and verifiable. Furthermore, the study’s reflections on the use of translators, interdisciplinary collaboration, and systemic challenges offer practical insights that can inform future outbreak preparedness.

Minor suggestion:

While the authors note that communication and visualisation are discussed in an adjacent study, a clearer reference (e.g. DOI or direct citation) would be useful for readers interested in exploring those aspects further.

Overall, this is a well-conducted study that makes a valuable contribution to global health systems research and policy preparedness.

Reviewer #2: The manuscript has complied with all the recommendations and made all necessary correction pointed out at the initial review activity. I hereby recommend the acceptance of the manuscript for publication on PLOS Global Public Health Journal.

7. PLOS authors have the option to publish the peer review history of their article (what does this mean? ). If published, this will include your full peer review and any attached files.

**Do you want your identity to be public for this peer review?** For information about this choice, including consent withdrawal, please see our Privacy Policy .

Reviewer #1: No

Reviewer #2: **Yes: ** Sikiru Olanrewaju Badaru
